# Revierparks as an integrated green network in Germany: An option for Amman?

**Maram Tawil** **1\*, Yasemin Utku²ᵒ, Kawthar Alrayyan¹ᵒ, Christa Reicher²ᵒ**

**1** Department of Architecture and Interior Architecture, School of Architecture and Built Environment, German Jordanian University, Amman, Jordan, **2** Department of Urban Design, Faculty of Spatial Planning, TU Dortmund, Dortmund, Germany

ᵒ These authors contributed equally to this work.
\* maram.tawil@gju.edu.jo

## Abstract

Amman city has gone through differentiated patterns and trends of sprawl in the last few decades. It went through disorganized developments due to reasons that vary from political, geopolitical, socio-economic and others. A threat that is existent in this perspective is the sprawl of cities and urban structures and the merging of them with no respect for the cultural and social needs of the community. Development of the city of Amman should take the shape of integrated planning that contributes to sustainable development. Taking into consideration the geographic layout of Amman and the tendencies of sprawl to overrun the green areas, the focus of the paper is directed towards the fringes of Amman and the seemingly available spaces of potential. A strategic approach for an open spaces network that allows for dynamic lifestyles in Amman is suggested. It should encompass a variety of safe and attractive spaces that are well distributed throughout neighbourhoods and accessible to the communities. Success factors of the Revierparks in the Ruhr area of Germany are researched in order to highlight potential strategic thinking in dealing with problems in Amman. Accordingly, defined magnets and anchor nodes in Amman are specified make the city more readable and accessible.

## Introduction

This paper is going to discuss the prevailing problem in the disorganized sprawl Amman city has been suffering from in the last few decades. During 1983 to 1996, rapid urban growth greatly increased the spatial area of Amman. In 1996, urban expansion was estimated to be 150,764 km² [1]. The paper will investigate options and strategies in containing the problem of overrunning green areas in the scene of such urban sprawl. Needed levels of interventions like recreational areas stand as crucial to enable interlocks between the urban structures or the amenity areas that can orient the community further within their city.

Planning of such spaces is vastly important to the community of Amman, as the city has grown according to many motivations including investment areas that are more residential and others; transportation networks, social infrastructure, amenities and open spaces were left

**Competing interests:** The authors have declared that no competing interests exist.

out in the process. According to Alnsour 2015, rapid urban growth, accompanied by insufficient legislation to control such growth and weaknesses in urban management, have created many problems in the urban development of Amman. Such problems include poor public transport and scarcity of green areas among other crucial aspects [2]. On another end, Amman is described as ever growing Amman, where multidisciplinary aspects are challenging the urban and regional development of Amman [3]. The neglect of these has resulted in a disorder of the different neighbourhoods and districts, the urban sprawl into the green and other aspects that can be considered as risking the open space and therewith the sustainable development of the city.

Development of the city of Amman should take the shape of integrated planning as well as sustainable growth strategies that ensure the responsiveness to all affected layers for development are considered. Taking into consideration the geographic layout of Amman and the tendencies of sprawl to overrun the green areas, the focus of the paper is directed towards the fringes of Amman and the seemingly available spaces of potential. The paper will discuss the options for development that occur while creating and sustaining a kind of a network of functional and recreational spaces as green open spaces.

Regional planning has much to do with the overall borders of cities and their geographic attachments to other surrounding cities. The discussion in this paper however relates more to the thinking of green areas that border and link the city to the region through incorporating the cultural need for space, therewith, the appreciation of the community of their city through the functional layout of such spaces. In this sense, a regional concept for Amman is needed.

Reflecting on this issue led us to study different strategies that have developed such networks; displaying a successful case of the Revierparks in the Ruhr area in Germany, and examining the successful elements of these regional parks as a concept of a recreational regional layout in addition to other functions within its development of cores. This concept in Germany has proven success since the 1970s and the paper will examine it as a model for Amman.

The objectives of the paper are to point out the direction of development needed in Amman to seek options and opportunities of adaptation of integrated Revierparks within open spaces in Amman. The paper is also striving for determining if the parks in Amman have integrated aspects that bring in a flow of users in addition to a flow of integrated land uses within and around them. Another objective of the paper is aimed at determining the needed functions shaping the parks in association with the raised local needs. Accordingly, a strategic approach in addressing the potential in Amman is aimed at. An approach that encourages dynamic lifestyles and provides with an open space network that is serving as a backbone for the whole city.

## Theoretical background

### Natural based solutions as an integrative pillar of development

Many societal challenges have been a major issue and a determinant for researchers and practitioners to seek solutions through time. These changes incorporate health, demographic change and wellbeing, food security, sustainable agriculture or climate change [4], and the absence of green infrastructure in terms of the urban setting—specifically the lack of urban open-space networks which are all needed for the functioning of a society.

Interdisciplinary sectors have adopted new solutions and concepts to address societal and economic challenges, called the Nature-based solutions (NbS). This kind of strategies can only be achieved through the influence of policies and institutions at all levels, which requires a network of connected protected areas covering the most important areas for biodiversity and integrated into healthy and resilient landscapes [5]. Moreover, it can take many forms and

typology, such as providing recreational and tourism benefits, developing green infrastructure in urban environments (e.g. green walls, roof gardens, street trees, or vegetated drainage basins) to improve air quality and conserving forests to support food and energy security.

It is suggested that NbS is an umbrella concept that offers nature-based solutions, ecosystem-based adaptation, green infrastructure and ecosystem services [6]. These Nature-based solutions are categorized into three different typologies: solutions that involve making better use of existing natural or protected ecosystems; solutions based on developing sustainable management protocols and procedures for managed or restored ecosystems; and solutions that involve creating new ecosystems [4].

Parks and open spaces as one type of green infrastructure, can serve as multifunctional nature-based solutions to achieve both climate-related and societal goals [5] in Europe. In this regard parks are seen as nature-based solutions for urban integration, municipal policy, and ecosystem services. The case of the Greenway Network Strategy for Lisbon established the base for Nbs to support deprived communities. Their main function is to serve food provision, e.g., for immigrants and struggling families, as well as cultural services, including recreation and a sense of place. The strategy of connecting these spaces to existing parks and therefore maintaining the existing permeable areas also contributes to improved green infrastructures by establishing ecological corridors in the city. Besides providing food, urban gardens contribute to creating opportunities for leisure and recreation and thereby promote health and well-being, as well as a sense of place, cultural identity, and social cohesion–important factors for societies to adapt to change.

## The need for public spaces in cities

Public spaces are widely defined within different settings and take on a variety of different forms. They can embody formal spaces perceived as centers for the settlement and the focus of cities and neighbourhoods where activities and public life take place [7]. According to James Mensch, a public space is a space where individuals see and are seen by others as they engage in public affairs. It refers to an area or a place that is open and accessible to all citizens where most events are spontaneous rather than pre-planned [8]. Furthermore, a public space is a place that has been reserved for the purpose of formal and informal sport and recreational activities [9], specifically designated for diverse active and passive recreation activities [10].

Public space provision in cities faces various complexities ranging from the impact of urban growth to the influence of other planning trends and spatial practices that are not tailored to the local context. On the one hand, urban growth in cities has increased the need to service delivery by accommodating traffic networks or new housing areas to meet growth. This has had an impact on urban landscape by causing a deep division and fragmenting the urban fabric of cities. On the other hand, in the case of Arab cities with emerging economy, the influence of western design criteria and spatial practices has affected the social divisions in the cities. It has caused discontinuity between old and modern cities, and complicated, and widened the social divisions in the cities [11]. Moreover, and according to different authors, public space is heavily affected by recent urbanization. Fragmentation of the urban fabric and deterioration of public space along with the privatization of public space are major trends in urban space transformation [12]. Open and green spaces are considered as a critical component of cities as they define the public realm by framing development within a network of parks, recreation areas and other open spaces that accommodate everyday social interaction. The planning of such urban green spaces increases the efficiency of green infrastructure that in turn intensifies and provides ecosystem services [13]. On a regional scale, open space systems connect cities to land outside of built-up areas providing opportunities for integration with surrounding

landscapes [14] but also contain growth and sprawl within the city borders through an environmental infrastructure in a sustainable manner and promote the concept of compact cities [15].

As a result, for the spatial and regional benefits of public spaces compared to the unprecedented population growth, the need for public spaces increased to support diverse recreational activities [16]. However, pressure for land becomes extreme as some cities do not have the means to cope with rapid growth, and therefore, the preservation of public space becomes a public burden that affects public life [17]. The benefits of a more compact city with more efficient growth through an urban green infrastructure are particularly evident in the case of Amman city that has an evident divide within its social structure, which is further reflected on the urban areas causing discontinuity and fragmentation.

## Integrated parks as a typology of public space

Due to increasing urbanization around the world, landscapes are experiencing profound changes in the provision of the needed green and open spaces [18]. The need for open space conferring with the daily life of communities is becoming a crucial need [19]. Spatial inconsistency between the supply of recreational services and the demand for them has raised inequity concerns regarding urban park policies. Need for recreation reached its peak after the industrial revolution in Germany as it led to profound social changes. Planners dealt more with the concept of recreation than leisure; they sought opportunities for activity after work, which refreshed and renewed the worker for more work and social matters. Revierparks is a leading new concept of parks that emerged in the 1970s as a post-war solution for the upgrade of the quality of life and the societal norms in the area.

New concepts and theories related to recreational parks have continued to emerge offering new strategies and solutions for urban and environmental developments. These will further have an impact on social, ecological, economic and regional layers in the cities. The above layers can best conceptualize strategic approaches that can integrate parks into the planning system and green network for a better sustainable development and quality of life [20].

On the social level, extensive literature assures that parks and open spaces are beneficial in many socio-cultural aspects, in addition to offering indirect health effects conveyed by providing opportunities for physical activity and by different modes of recreation [21]. For that reason, a recreational park should facilitate minimum requirements and space for gatherings, meetings, recreational activities that promote social interaction; such as playing, exercising and picnicking. Those activities are offered to address the needs of health and social cohesion.

A study conducted in Helsinki, Finland, indicated that nearly all (97%) city residents participate in some outdoor recreation during the year [20], which means that people are encouraged to participate in such activities if provided. Another study conducted in Sweden, found that people who were exposed to the natural environment are less stressed compared to people who were exposed to the urban environment. This indicates that urban green spaces have a positive impact on physical health and wellbeing [22], and serve as a resource for relaxation and emotional warmth [20]. When people are engaged in activities that connect them, it enhances the social cohesion.

Economically, urban public spaces and parks can cater for economic development as the quality of place, accessibility and multi activity can lead to a user's attachment to the place and the economic activities surrounding them [14]. Additionally, indicators are very strong that green spaces and landscaping increase property values and financial returns for land developers, of between 5% and 15% [23]. This should encourage investors, policy makers and developers to invest in more economic developments within these open spaces and parks.

Various approaches consider providing economic opportunities benefiting the parks and public spaces to help in maintaining them. Generating economic opportunities takes place by re-activating new functions and recreational activities of existing parks themselves. Such activities can mean incorporating social groups that are willing to pay for the use of such activities such as swimming pools, a gym, mini-golf courses, indoor playgrounds and restaurants, etc. . . . Through such activities, municipalities can sustain maintenance and management of these parks. These functions could be part of the development of recreational parks to serve as a catalyst of change or as engines that generate economic opportunities with different target groups as high quality public and open spaces can bring in great revenues to the surrounding neighbourhoods and urban settings on economic, social and environmental levels [7]. Furthermore, integrated approaches to regional parks contribute to creating nodes with the surrounding smaller areas that exist as anchors, with certain identities, for economic development, as is the case in Amman [24].

Environmentally, urban public parks and open spaces can provide benefits to the environment and urban climate. They not only decrease the urban heat island effect caused by heat-absorbing surfaces in cities, but also preserve the biodiversity and optimizes natural ecological system [21]. They offer appealing views, can help to clean the air, reduce noise, control pollution and have positive microclimate effects [25]. Such benefits can be obtained by the large size of green space and vegetated parks balancing some climatic inputs such as; solar radiation, relative humidity, air temperature, wind speed and relative humidity. A study in Chicago has shown that increasing tree cover in the city by 10% may reduce the total energy for heating and cooling by 5 to 10% [23].

Spatially, green open spaces and parks should be planned in the right location. They should be accessible and functional, where accessibility and distance are a measure of a park's ability to provide services [26]. Different spatial scales from the neighbourhood, city, sub-regional and regional levels are needed to guarantee a proper allocation combining appropriate types and quantities of components of recreational facilities [27]. The article discusses the existing parks in Amman that can be relevant to the discussion. In this sense, the size of the parks and the readiness of functions relating to different users are in many ways measures to attract distance visitors [28]. However, distant visitors are dependent on a sound transportation system that enables easy accessibility. Therefore, parks on regional fringes can have this benefit to development as they release condensed large cities [29], yet are reliant on associated measures accompanying the planning of such networks like infrastructure, services and spatial organization of the surrounding urban space [28].

Land protection in this sense is another benefit of securing a green network, which not only improves the ecological environment of the city region but also provides important support to urban environmental improvement and nature conservation [30]. Urban green spaces provide the linkage of the urban and rural areas [31]. A functional network of green spaces is important for the maintenance of ecological aspects of a sustainable urban landscape [32].

Regionally, green space gives the city a coherent structure to connect different scales and parts of the urban fabric through urban and regional parks. Various benefits are attainable from the regional network of public space. Creating green network connected within visual or physical spines would create a tissue of cohesion. Green belts, for instance, aim at preventing urban sprawl towards surrounding urbanized areas. Urban and regional parks' planning and design should aim at producing spaces which are attractive and accessible to people. Different models can be adopted to bring in the best use of parks on the fringes of cities as well as within their urban fabric [33].

## Case analysis of Revierparks in the Ruhr Area: More than just a park

The Revierparks were built in the Ruhr area in the 1970s. The Revierparks introduced a new type of park to the region of the Ruhr that combined and integrated multitude dimensional functions of a public space within their locations. They were integrated into a regional leisure concept, which was developed in the context of a plan for the planning of the Ruhrkohlenbezirk (The Ruhr Coal District) (SVR), now the Regional Association of Ruhr (RVR). The district parks were an essential building block in the new programming of the industrial landscape, which was made more attractive by the "Development Program Ruhr" (1968), with the statement, "The industrialized landscape of the Ruhr region that has to be made more attractive by the expansion of further regional recreation facilities" [34]. Since 1970, five district parks have been created with areas of between 30 and 40 hectares, plus adjacent free spaces connected to the parking areas. The parks themselves have a wide range of open spaces and recreational themes in high density and complexity that are interlinked and interwoven so that they act like "landscape machines". The district parks were a model for recreational development in the Ruhr area and an integral part of the leisure activities of the young and old in the region.

In their basic function, the Revierparks were recreational destinations. They have since expanded their role in that they now form a social backbone and a structure for the regional layout of the area. The Revierparks concept was also adapted to address the new needs and challenges of the area, yet have not compromised their original concept. Furthermore, the Revierparks still cater to their original purpose as recreational destinations. Furthermore, they have yet developed their role as being a main contributor to environmental and social construction for the whole region., and in fact have not limited their role to the regional layout, but also engaged in integrating functions that have link to the neighbourhoods in their proximity. The plans for the Ruhr area, which were submitted by the SVR in the post-war period, were associated with major objectives. These included the separation of functions and secure spaces for infrastructures. Other objectives dealt with issues such as open space and environmental protection as well as the establishment of recreation areas on the regional level [35].

As shown in Fig 1, the sites of the regional parks have conceptually linked and secured the regional green areas in the Ruhr region. They have also ensured the link to the network of green spaces between the centres of the settlements to enable the social life through the nodes they are creating with the regional network. The selection of the locations tended to be on formerly agricultural or pre-industrialized areas or open spaces. Each district park should be accessible to the residents of at least four surrounding routes in a reasonable travel period [36].

### Functions and characteristics of revierparks

The Revierparks had a focus on three main elements that reflect the programme and role of the parks in the different areas. These elements were diverse in their function to meet the different needs of the target users such as the recreation seekers using the parks after work or even those having longer visits through weekends. The first of these elements include sports playgrounds, social interactive spaces and children's playgrounds. The activities that make up this element are always free of charge and people can use them without any limitations. The second element comprises of adjacent natural green spaces to the neighbouring settlements as well as to the social activities created in the park. These types aim at creating and offering a link to nature, enabling long distance walks, natural outings and quiet relaxation in a nearby destination. The third and final element involves financial investment. These in turn add the flavour of extra activities and exercises to the parks and are not free of charge. They could consist of private investments that take the form of health centres, restaurants and shops or even

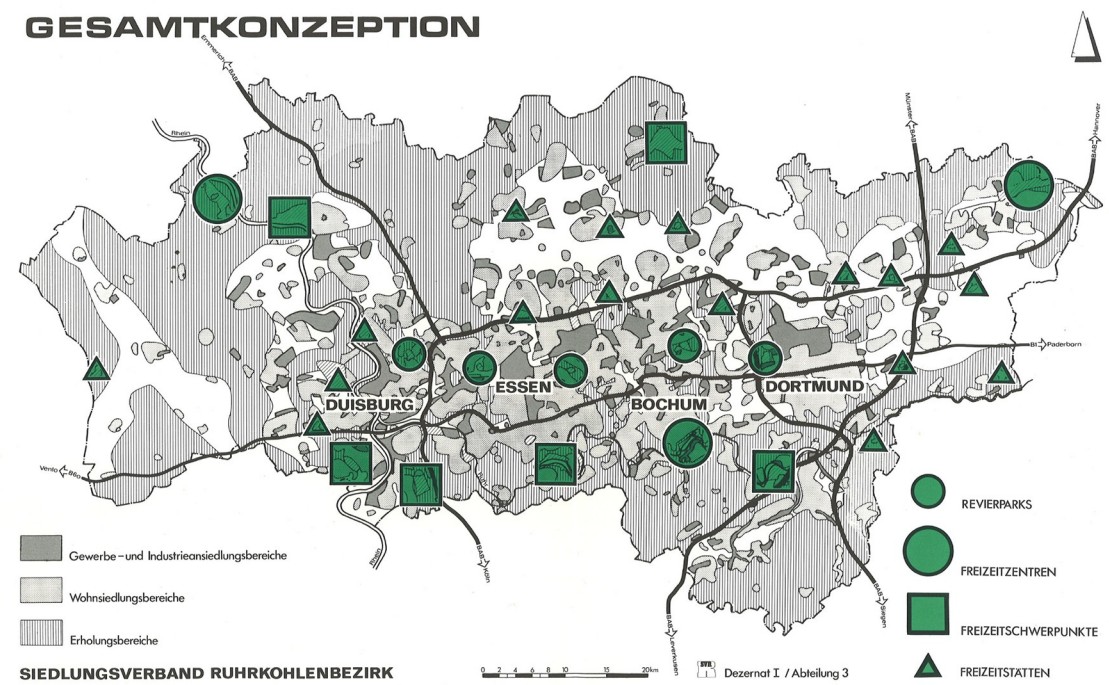

**Fig 1. General concept of the parks in the Ruhr Area, Germany.** Source: SVR–Siedlungsverband Ruhrkohlenbezirk (Hg) oJ, S. 17.

luxurious sports activities like mini golf and ice-skating in some parks. The aim of these types is derived from the intention of engaging all social groups into the situated parks and encouraging them to see the potential of using the free spaces while visiting the private ones as a synergy to activate the communities in differentiated venues in one.

The timeline of establishing these parks started with the Gysenberg Park: the first, largely finished park, which was opened in Herne in 1970. In 1972, the Nienhausen area between Essen and Gelsenkirchen was established, followed by the Vonderort Park in 1974 between Bottrop and Oberhausen. The Dortmund Park in Wischlingen was established in 1976, and the Mattlerbusch area in Duisburg, completed the entire "Revierpark" project in 1979. The distance between the parks is about 10–15 kilometres. The individual sites connect through already existing "landscapes" or recreational facilities.

The parks are governed by the neighbouring municipalities on one side and the central authority for governing the green space in the whole Ruhr region (RVR) on the other. Normally, each authority governs 50% of the RevierparksThe aims of this decentralization and yet central governance is to ensure they are run well and maintained for the communities around and within the municipalities. These different governance entities are described in Table 1 for the five locations.

The Revierpark Wischlingen can be studied as a typical example of the Revierparks of the Ruhr region in its function and layout. The spatial concept of the park is divided into the three parts. The first is the commercial part with indoor and outdoor activities. These comprise of swimming pools, health centres and recreational activities along with various restaurants and gastronomies. This first part, which forms the attraction for the visitors aiming at a day recreation, is always concentrated in a cluster in the central entrance of the park. Attached to it is the second part with its green spaces which forms a circulating layout that allows the communities to visit and enjoy their time in a free manner without having to pay charges. These mostly take

Table 1. *The five revierparks and their governing municipalities in the Ruhr Area.*

|  | Gysenberg | Nienhausen | Vonderort | Wischlingen | Mattlerbusch |
|---|---|---|---|---|---|
| **Opening** | 1970 | 1972 | 1974 | 1976 | 1979 |
| **Size** | 31 ha | 38 ha | 32 ha | 39 ha | 40 ha |
| **50% RVR** | Municipality Herne 50% | Municipality Gelsenkirchen 25% | Municipality Oberhausen 25% Municipality Bottrop 25% | Municipality Dortmund 50% | Municipality Duisburg 50% |
|  |  | Municipality Essen 25% |  |  |  |

Source: Regionalverband Ruhr (Hg.) 2015; modified by authors)

place in adjacency to historical buildings or sites that invite visitors to come and enjoy the views they offer. The different facilities and functions of the park are always changing in accordance to the needs of the cities and communities. They also change according to investment opportunities. Yet, the form and layout of the park stays sustained. The Revierparks have always many entry points that allow neighbourhoods and communities to approach them from every direction.

## Benefits of a Revierpark: Social, economic, environmental and regional impact

The Revierparks are more than a normal classical park; rather this type of park combine and synergize social, economic, environmental and regional aspects in one framework, which makes the park function as a development anchor for the setting.

In terms of social aspects, the Revierparks are destinations for recreation and relaxation within the everyday life of the communities. At the same time, Revierparks offer a wide range of activities to the visitors that vary from a place to go for picnics with family and friends to spaces for hosting big concerts and activities where the public can attend free of charge. They also include spaces for different extra curricula activities that take place outside of school hours for children and youth [37]. Revierparks are hence, a place for associations and organizations to practice their events and activities in public and in the outdoors without any charges. This type of utilization of space caters to social inclusion and social equity and stands as a contributor to the development of the neighbourhoods around them and therefore to the development of the whole region. In this sense, the accessibility to the parks and to green is a major attribute to the success of such parks. Revierparks are destinations that can be visited by foot or using public transport from the neighbouring areas [38]. However, other studies and researches, have shown this is not always the reality; the majority of the Revierparks users, along with those to other recreational destinations, are making their visits using private cars [39], which is an issue that calls for a better-integrated system that should serve recreational activities.

The economic aspects are considered the most important factors of success in Revierparks. The different private activities running in the parks such as health and recreational centres serve as the engines through which the parks are sustained and further developed. They also cater for a regional framework, where the parks take the lead in steering the development of that framework. Hence, regions are being acknowledged through the associated parks. Economically, parks have certain financial gaps to cover their expenses. This is well known in the articulation of open spaces [10]. Therefore, Revierparks have intervened in the sense of accommodating commercial facilities and functions to bridge these gaps and bring more investment on certain land use within the parks.

Environmental aspects are of key importance to the concept of the Revierparks, which underlines the compatibility between the recreational space and the preservation of green.

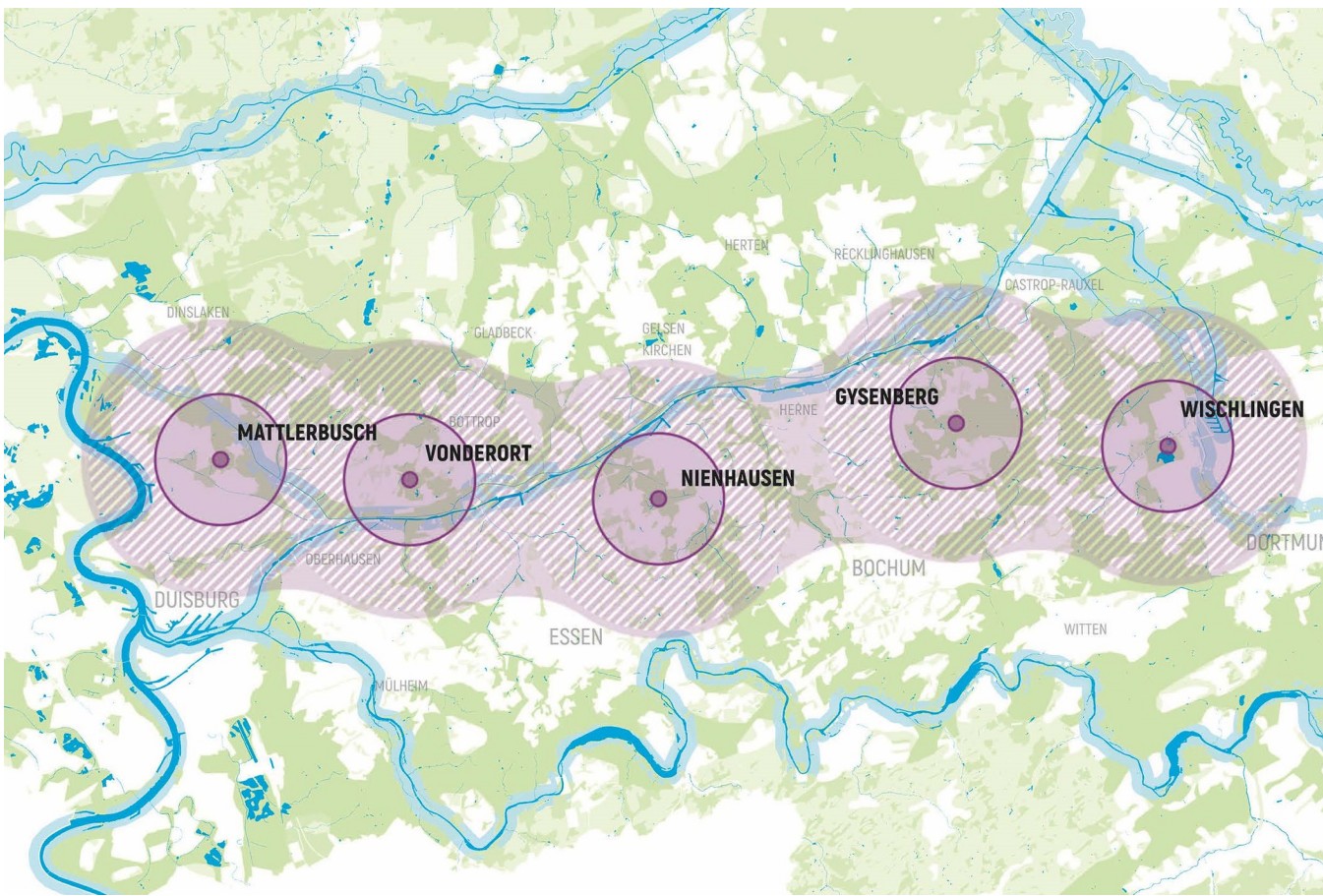

**Fig 2. Network of revierparks in the Ruhr Area.** Source: RVR (2017) RHA, PGO, sds_utku, 2017.

This synergized concept has won weight nowadays in the different settings. The Revierparks are a live example of such integrated development and have proven to be products of well thought out planning, effectively articulated, where climatic and environmental functions along with recreational services serve the communities and the development of preserving green in cities and regions.

On the regional level, Revierparks are working as a network within the region and between cities in the Ruhr region. They serve as meeting and starting points within an outer line for cities that connect the areas for regional tours and along a nature-based open space network [37]. They are 10-15km apart and create a belt of green areas around the cities but also are a line that offers multitude dimensional functions and caters for a recreational policy for the cities as in Fig 2.

## Amman as a research setting: Methods and data collection

Amman is a city with a population of 4 million inhabitants within the boundary of the municipality [40]. The population has grown radically from 1924 and the rate of urbanization has increased from 44% in 1961 to 83.6% in 2013 [40] as a result of many incidents, such as; wars and crises that affected the region causing of successive waves of refugee migration, from Palestine, Iraq, and Syria [2]. Moreover, the stable economic market and improved health care among the unstable situations of the region has led to most of the numerous waves of

displacement and refugee migration [41]. Topographically, which consisted of steep hills and narrow valleys, had a major role in shaping the urban sprawl of the city and its flow of expansion through the years. One of the main directions of the rapid growth is towards the northeastern industrial town of Zarqa and the south-east (where 80% of all industrial services are to be found in Amman) [42].

This rapid growth has been the main reason for the social divide that has come to characterize the residential areas of present-day Amman; characterizing the urban public space of the city that has been transformed within the urban area, stretching public services and open spaces to the limit. The social transformation of the city of Amman has been no less spectacular, as it has suffered from the continuous segregation due to the unplanned distribution to social groups within areas of services that have divided the city into two parts; west and east Amman [14]. This east-west division has grown along social and economic lines measured by levels of education and literacy, occupation and employment, housing type, and income [43]. The eastern part, which includes most of the informal settlements that developed following the arrival of the Palestinian displaced persons from the West Bank and Gaza in 1967 [44], lacks public spaces the most.

The continuous urbanization and influx of refugees have created numerous challenges, such as informal settlements, overcrowding, and degradation of agricultural land. Hence, there is a lack of open and green spaces with low quality urban services [2]. Over the past few decades, a large number of green areas have been converted into construction projects; for example, the Amman Gate Towers project on top of an urban park that aimed to serve the residents [43]. As a result, urban parks are left with the remaining vacant gaps, so that Amman has now uneven distribution of parks today. Therefore, if the current trend of urban growth continues into the future, the city of Amman will face many environmental and urban planning problems due to further urbanization. Therefore, alternative scenarios for the sustainable development of Amman and Jordan are required. In addition to Planning, strategies to connect the green to the existing potential open areas and suggested ones to contain the growth and orient it to certain spines is urgently needed.

Greater Amman Municipality currently has no designated Open Spaces System and therefore is unable to define connections within and between Settlement Areas within its boundary. This is reflected in the void left out within parks and public areas in the built-up portions of the city. Focusing in the east of Amman, where the population density is the highest, many parts of the city do not have open spaces except within scattered, private, vacant lots; mostly public interaction occurs entirely on the street and in places of worship.

Previously, the 1988 Greater Amman Comprehensive Development Plan included many policies related to the natural environment and open spaces, where one of the goals of the plan was to develop an overall open spaces strategy that would include parks, forests, recreation areas and agricultural lands within an overall system. Unfortunately, the 1988 plan did not take concrete steps towards implementation. The Open Spaces Plan renews the earlier 1988 plan concept by identifying an Open Space System for Amman as it currently has a number of parks and forest areas designated within the Urban Envelope but not connected within an overall system.

Looking in depth within the existing urban fabric, a number of spatial nodes stand out in terms of being potential jobs generators in Amman, and creating major open spaces to target, such as Marka and Sahab in the northern and eastern part of Amman, and in the southeast, where the major population concentrates- al-Qwaismeh and al-Moqableen areas stand out. One of the most important obstacles facing the public sector in many developing countries is its ability to improve the quality of life, provide effective urban services, and raise living standards under the severe challenges of rapid urban growth [2].

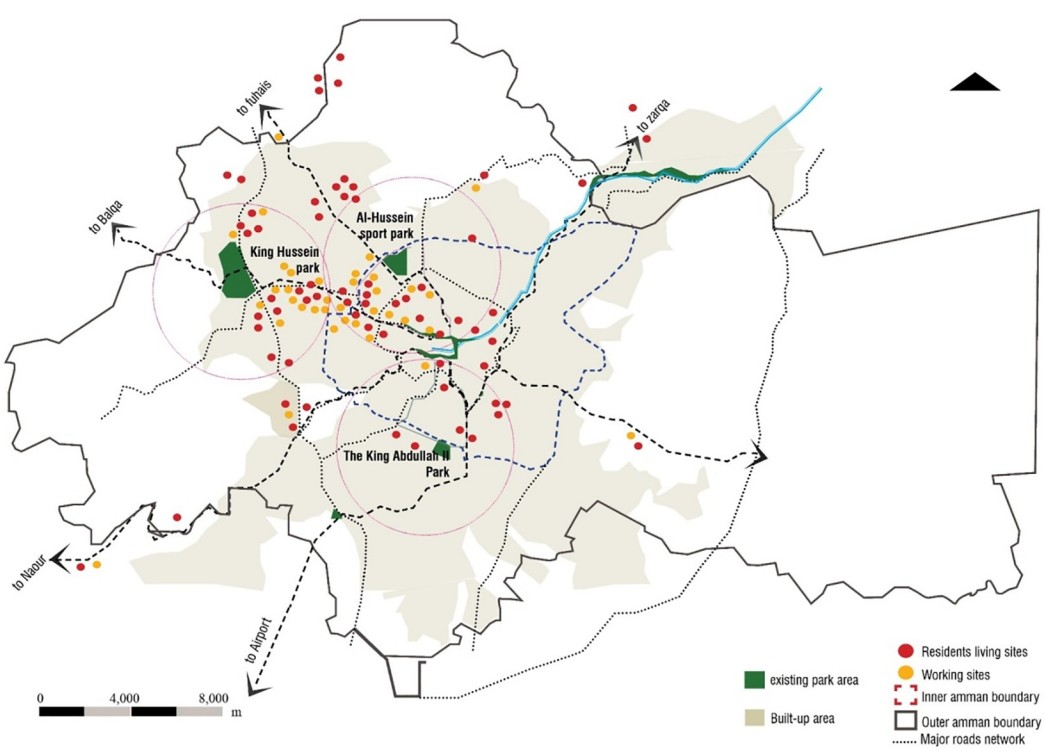

**Fig 3. Survey samples mapped in Amman.** Source: authors 2018

The three main developed destination parks in Amman that were created on a large scale were analyzed to give a better insight into the public green space quality of life in Amman. The aim of creating these parks on the part of the Greater Amman Municipality was to provide the local community with recreational space and green nodes in the different locations in Amman as shown in Fig 3. In principle, these parks demonstrate the need for green and open spaces within these areas, as it is required for any community in an urban setting. Their aim was to target all social groups at different times of the day and in a frequent manner, that promotes a healthy quality of life. Such interventions were intended in principle. In practice however, they have shown different results that underscored the challenges outlined previously regarding planning gaps in Amman.

The aim of the survey that targeted 200 persons and was distributed around the selected parks as shown in Fig 4 was to point out the direction of development of such parks and main open spaces in Amman. The objectives of the survey were mainly to determine if the parks in Amman have integrated aspects that bring in a flow of users in addition to a flow of integrated land uses within and around them. A further objective is to identify the link and relation between the different parks. These objectives were examined through investigating the reasons for visiting these parks, and defining the gaps in their efficient use. In order to meet these objectives, the survey was conducted to acquire information to identify if the parks meet the needs of the users, are adequate for the communities around them, are accessible by the different social groups and if the parks need further developments from the user perspective to reach the level needed. The data collection process also served the objective of determining if the location and functions of the parks enhance development aspects like social inclusion and economic development.

The methodology in this sense has taken two pathways. The first one is analyzing the five Revierparks in the Ruhr Area in Germany with the support and the approval of the Urban

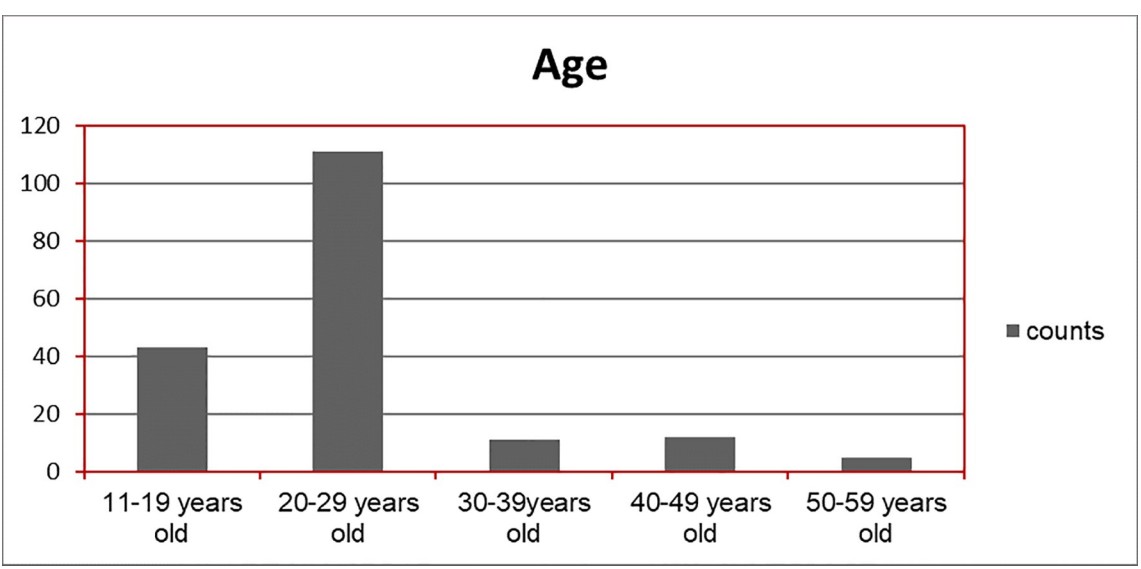

**Fig 4. Parks users in Amman according to age groups.** Source: authors 2018

Design and Land use planning Department at the TU Dortmund in Germany that has done much work investigating the development tracks of these parks. The survey of this first pathway has taken the shape of observation and documentation of the different behaviours in the parks, as well as their accessibility and uses of functions at different times of the day. The documentation process undertaken in the revierparks formed the base for building the measures for the further survey. The successful aspects of the revierparks are tested in the local setting of Amman to seek representing factors enabling parks from attracting and receiving more target groups. A merge and synergy of different activities are also tackled to see what options of such interactive spaces are valid for the communities in Amman.

To address the objectives of the research, a questionnaire was developed to build on the former analysis of the revierparks and bring in evident measures to be undertaken and adopted in the further planning of parks in Amman. The questionnaire is designed to investigate the status of parks in Amman, and the frequency of visitors as a first step. It tests the missing functions that prevent the local parks from fulfilling their role within the city and then identifies the needed functions related to the different target groups addressed in the survey.

This forms the second pathway of the methodology and was distributed anonymously in the selected parks in Amman with the aim to document the real use of the parks in Amman and the type and frequency of users. The fieldwork conducted in Amman was within the framework of the cooperation project and approved by both universities and the Scientific Research Board.

The sampling of the survey was stratified to have all possible users in the parks, according to gender and age. A random sampling procedure was adopted to create a representative base for the park users and enable the research to come up with reality-oriented results that can serve for a better framework of park and network development [45]. The sampling size was 200 respondents randomly selected to target the users of the parks. It was noticed that the parks are not visited in big numbers, therefore, the survey took place in different times of the day and different days of the week to capture the reasons behind using or not using the parks or to capture the well represented opinions of how to develop them in a better way.

As for the target groups and users of the different parks, the results of the questionnaires show that the parks have succeeded in attracting certain age and social groups out of the whole

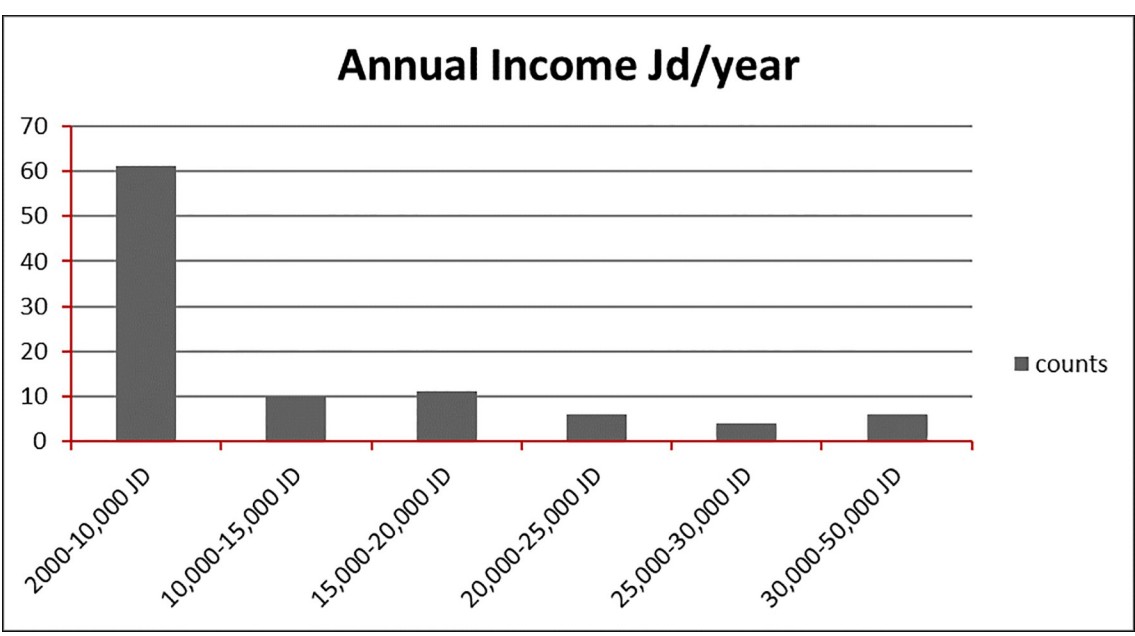

**Fig 5. Parks users in Amman according to Annual Income.** Source: authors 2018

community. As shown in Fig 4 and Fig 5, the age groups using the parks are limited to between 17 years old and 27 years old. Whereas, the social groups visiting the parks range in their economic status between low income and middle income groups as shown in Fig 5.

It is worth noting that the households of the visiting groups of the parks have an average household of 4–6 members. This distinguishes the type of community using the park. Having mostly low-income social groups using parks indicates the notion of social segregation in the community in Amman. Mostly better income groups are more attached to functions and activities of social appearance within better contexts reflecting their economic or social status. The parks in Amman in their current developmental state have failed to serve their aim and do not contribute to a more integrated setting. Reasons behind this notion can be, in relation to the success of the Revierparks, due to the functions of the parks. The facilities and activities experienced at the Revierparks, as well as even to their layout and the connection they provide, address the requirements that arise due to the daily activities of the different social groups they cater to. This can be evident in the case study below that gives an example of successful links to the communities through an integrated park network.

The survey of the Amman parks also reveals the fact that even the groups using the parks more frequently than others have an average use of only five times a year as shown in Fig 6 or once or twice a month to the maximum.

On the other hand, more investigation has targeted the driving forces that would encourage more frequent use of the parks. Many reasons suggest economic facilities and commercial and business functions are seen as a direction that park development should take, as is seen in Fig 7. Fig 8 offers and suggests further activities and functions requested by the community to have the parks more as part of their daily lives.

Parks in Amman have to develop according to differentiated aspects rather than limiting its planning to conventional functions. They possess other potential that enables them to be part of the structure and the use of the whole city, especially the areas around them.

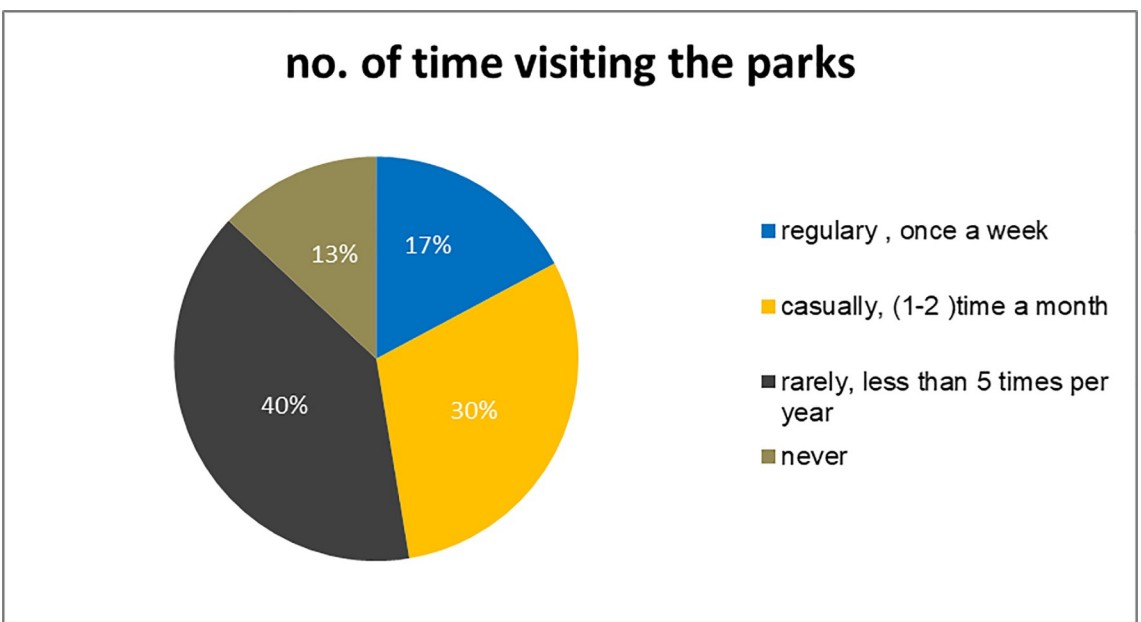

**Fig 6. Frequency of using parks in Amman.** Source: authors 2018

## Discussion

According to the data collected in the surveys, it is argued that social, economic, environmental and regional aspects are unnoticed in the development of public green space in Amman. Therefore, many problems in terms of the different layers have occurred in the setting of Amman. Fig 9 shows a comparison between international discussion on parks, specifically the Revierparks concept and its implications, and the situation in Amman, and in turn, offers a reflection of the key measures needed in Amman.

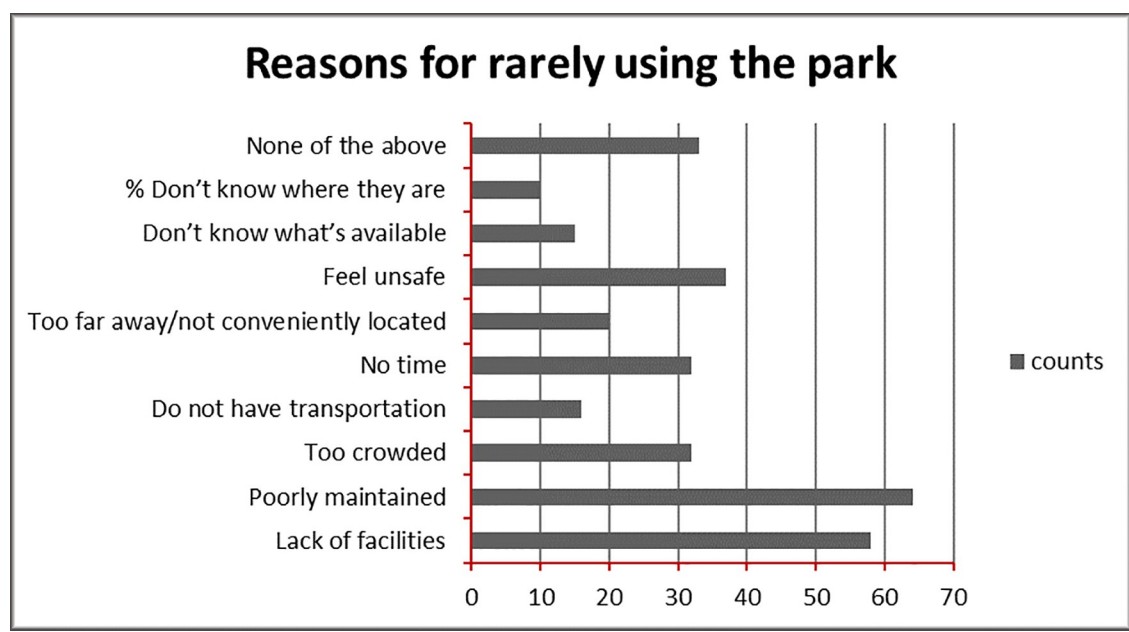

**Fig 7. Reasons for not using parks in Amman.** Source: authors 2018

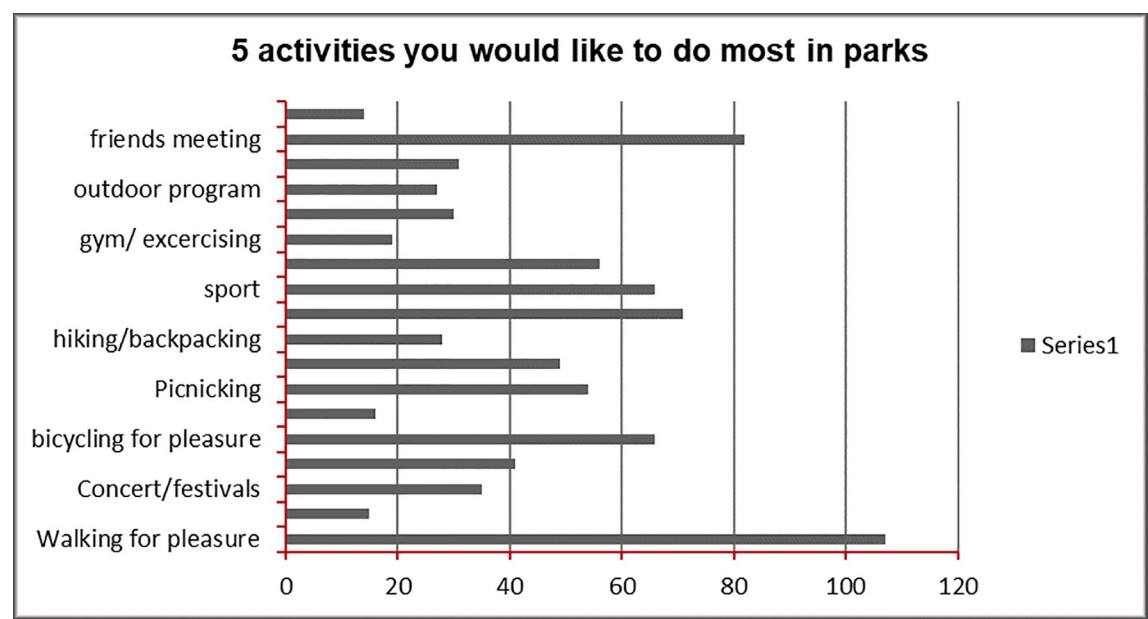

**Fig 8. Suggested activities by the local community to improve parks situation in Amman.** Source: authors 2018

From the social point of view, social inclusion is a major need in reference to existing social polarization in Amman [41]. Social exclusion is visible in many dimensions. Public spaces in west Amman are not welcoming to other groups from lower income areas. This is shown through the level of income of social groups visiting the different parks according to the survey. It is also evident through the lower income groups' behaviours at new projects and markets in Amman. The survey shows how certain parks are welcoming specific social groups through their structure, type and cost of activities within their premises. Such parks tend to keep their standards by indirectly excluding other types of social groups, so are located in better off areas in Amman and have functions that cater for high-income people. According to the survey, the public parks within the areas of east and middle of Amman are attracting all social groups. However, do not possess the needed activities and the variety of functions that keep their standards and exceeds their revenue to be able to maintain themselves, although socially they have potential. This resulting aspect goes in line with previous discussion in the literature as stated by [7], that generating economic opportunities takes place by re-activating new functions and recreational activities of existing parks. Such activities can mean incorporating social groups that are willing to pay for their recreational time. Hence, municipalities can sustain maintenance and management of these parks. These functions could be part of the development of recreational parks to serve as a catalyst of change or as engines that generate economic opportunities.

The parks and public spaces in Amman lack the sufficient revenue that maintains the structure and can keep them in use and function. Parks are constructed in many places and are developed and equipped in many areas. However, they are injected with these investments for a certain time and to cater to a particular community. They do not cater for broader uses and wider social groups. Parks in Amman are either located in certain areas where they function as neighbourhood parks for the surrounding communities, or are constructed in a place that cater for family picnics in the natural environment. They do not cater for multiple uses and broader target groups that enable continuous development. In addition, they rarely encompass the type of integrative uses like in the Revierparks that fosters development. In order to enable

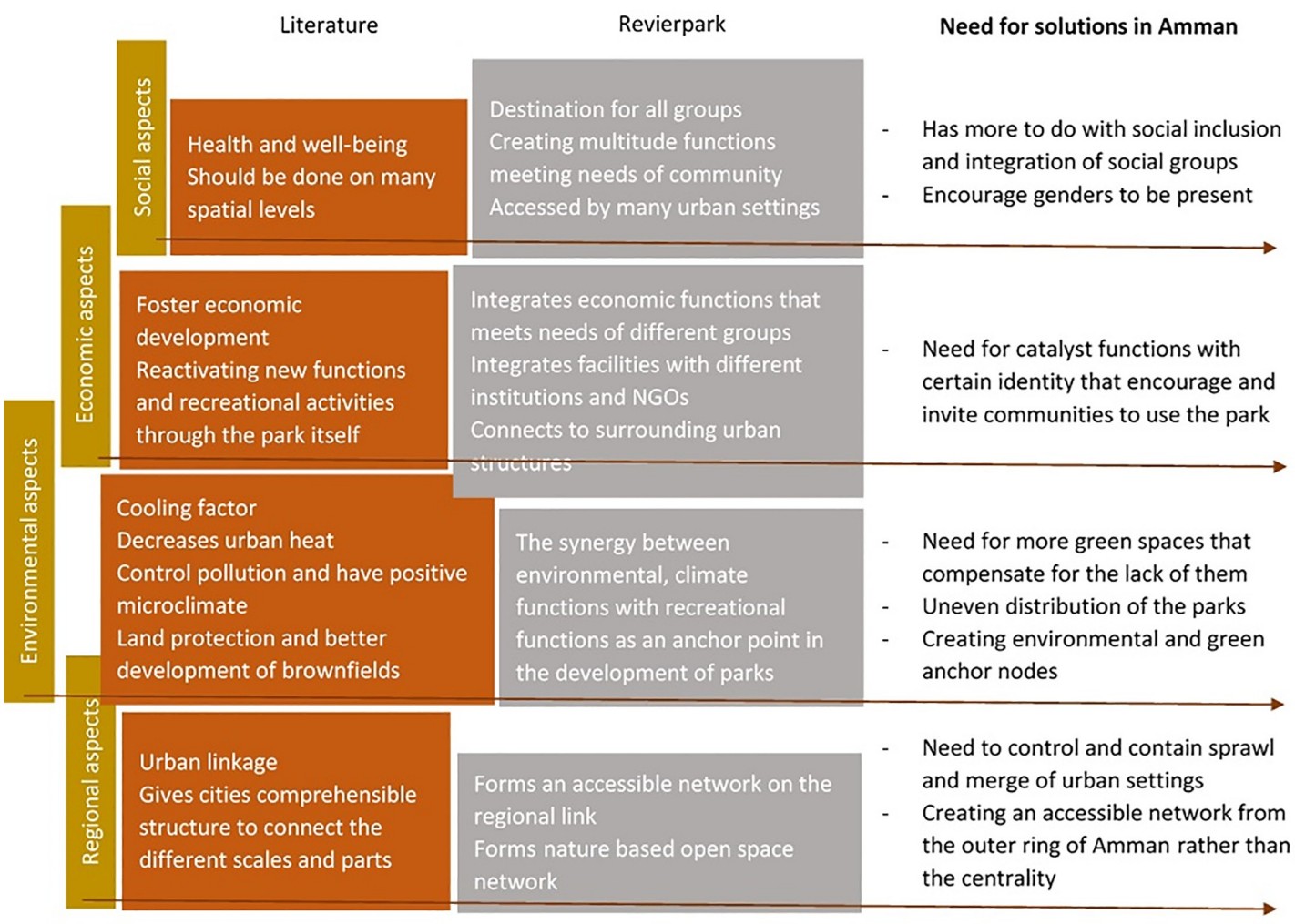

**Fig 9. Comparison between literature, revierparks and Amman.** Source: Authors 2018

such a development, more catalyst activities should be constructed and engaged to provide for a wider network and be in function with existing communities for differentiated reasons.

Environmentally, open and public spaces in Amman are constructed in areas of need within the neighbourhoods, or a bit further to establish a destination park for instance. These parks were constructed during the urban sprawl of Amman that defeated the green areas around the city through an unorganized sprawl. However, green areas are still to be spotted around the urban envelop of Amman. These are preserved in some parts to serve as regional parks and are considered important for the wellbeing of the community. While Amman is further facing and experiencing this type of disorganized sprawl, the danger of approaching these existing green areas and regional parks by the newly planned and built residential areas is still present. Regarding the inner parks in the urban setting of Amman, they have also to be linked through liveable corridors whether ecological or integrated with different land uses that correspond to the needs of the communities in the different areas. This is evident in the Revierparks, where part of their success that they are connected to an access throughout the city and not a stand-alone destination. It also stresses the issue of having parks accessible and functional, where accessibility and distance are a measure of a park's ability to provide services [26]. On the other hand and according to Wenping 2017, the size of the parks and the readiness of functions relating to different users are in many

ways measures to attract distance visitors [28]. Effective integrated development of the revierparks has ensured the preservation of the green, yet within an action framework that enables investment of a potential hub that caters and serves differentiated purposes, whether environmental, social or economic. Such a development motivates the effectiveness of the approach in the case of Amman, as preserving the green is not a competing strategy if seen as an investment in the scene of the promoted and approved urban changes happening in Amman. This links to the innovative frameworks created for the revierparks, the thing that is not as easy in Amman, but can be replaced with a hub around the park and in proximity to the place where different facilities, businesses and economic center is articulated that can cater for new and different use of the park. On another level, land protection is a point that could regulate and conserve the green spaces within a functional network in the city [32]. This matter is important for the maintenance of ecological aspects of a sustainable urban landscape that is contradicting to the case of Amman. However, such interventions raised from the survey in the parks can substitute for an urban policy in the regard and encourage it for the future development. Another benefit of securing a green network, which not only improves the ecological environment of the city region but also provides important support to urban environmental improvement and nature conservation [30]. The need for green space was proven in the survey, where parks are perceived as important on every level but not accessed on daily base.

Tailoring the approach within the context of Amman also encourages promoting a nodal network based on the green areas, which as a result can target different users to visit the green and open spaces on the one hand, but also indirectly encourages them to visit them while heading and targeting other uses in the vicinity.

In addition, while observing and surveying the five Revierparks in the Ruhr region, many aspects were visible to make the Revierparks different from the concept of a normal park. The survey was conducted through site visits to the five Revierparks. Observations and interviews to the local community using the parks were made to build a picture of how they are used, at which times and for which activities. The site visits were made at different times of the day and week in order to be able to analyse the impact of the park on the local communities, including even at working hours during the week to identify the needed activities for local users whether on weekends or after work. As shown in Fig 10, the integration between green open spaces and the collection of various centres that cater for different activities is evident when using the Revierpark. On the other hand, the parks engage temporal activities of different organizations and provide the space for those activities. Such activities sometimes include concerts, and at other times include workshops of a certain organization, and so on. In this sense, Revierparks possess an extra value over the normal public spaces and parks in cities, as people visit the park for recreation, work activities, outings and family picnics in addition to different programmes embedded accommodation and overnight stays. The issue of gathering these different types of users enhances and builds the image of the park and makes it more attractive to receive more and longer visits at a time. Further to that, research on Revierparks in Germany indicated the value of having catalyst activities to increase the quality use of the parks and bringing in more target groups that would not otherwise visit the parks. To this end, the Revierparks in Germany have developed in a competitive manner to win more target groups even from remote areas to use the offered activities and functions in the one park from another.

The successes of the Revierparks are tested in accordance with the deficiencies in green infrastructure in Amman. The aim of the article is not to transform the conventional parks or green open spaces into another image. Rather, they should accommodate differentiated and innovative functions, have the right location within neighbouring urban settings and have the potential of integrated land use that enables visitors to be in the park but also indirectly using the park while targeting other functions.

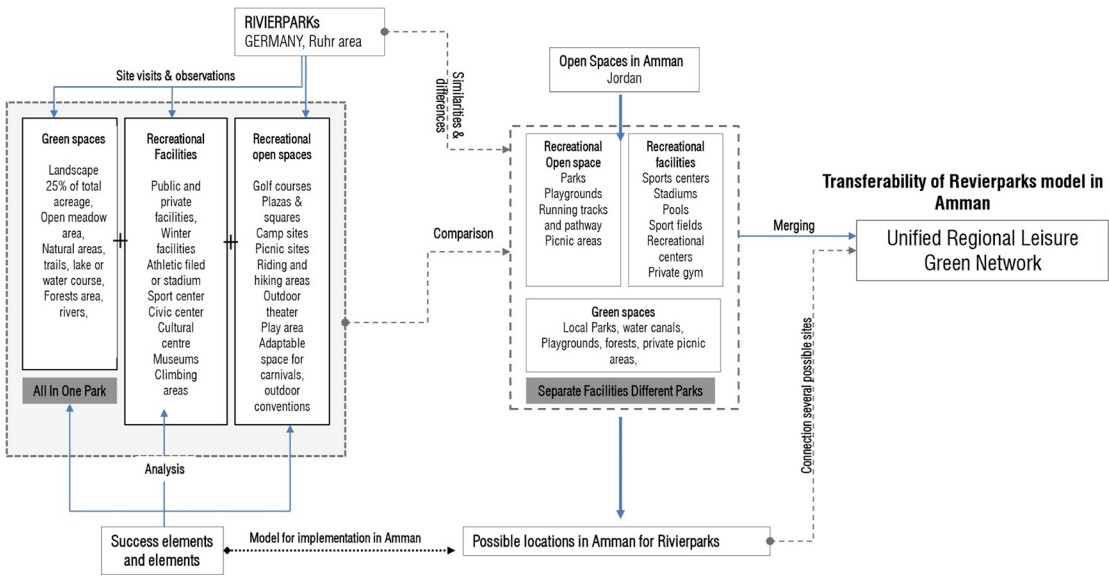

**Fig 10. Synergy effect between conventional and revier-parks.** Source: Authors 2018

Conclusively, parks can adopt different themes and innovative functions according to the identity of the area or the park itself. The functionality can strive for sizing the park's own potential and building on its assets.

Through analysing the Revierparks, it was evident that specific functional potential could arise. Laboratory for innovation was one of the themes in one of the Revierparks. Accordingly, the plan was made to relate the park to scientific activities, after school and after work workshops and to be a space for community gatherings for official reasons and sessions. This could bring in a new flavour and therewith, new target groups leading to a refreshing atmosphere and activation of the site. Identifying the Revierparks with the different identities and functions promotes also a balance in users using the one park from another without causing an unneeded competitive atmosphere between them.

The Revierparks have gained in importance as a result of their location, proximity to similar or related entities and land uses and the types of communities adjacent to the park. Such themes were for instance, the mechanical-physical laboratory, giving the opportunity to conduct experiments in parks, with historical settings, thus allowing for using this natural richness in scientific research. Another theme took the direction of a food laboratory. This park was adjacent to agricultural land that enabled an idea to develop a new type of recreation for children and families through natural farms and different associated functions. Other themes like water, repair laboratory and natural reserve laboratory were injected. In this sense, branding the Revierparks can take place according to the potential layouts or adjacent assets that can steer the development of such parks.

In the case of Amman, the city generally lacks the vital spaces needed that would allow for targeted functions and in turn such varied user experience. Therefore, the article will address the success factors that would be feasible in the case of the city of Amman to enable the development of an open green network.

## Towards an open green network in Amman

A vision of urban nodes constructed around existing natural spots can contribute to the network of green areas necessary for Amman. It is consequent that this network would provide

the city with the needed multitude-layered development around these nodes and form a network of major spaces.

Resulting measures to identify the locations for such development are based on the case analysis of the Revierparks and the survey highlighting deficiencies in Amman open spaces. The main characteristics in the existing green were; the size of a minimum of 30h, accessibility by main roads and regional surroundings, adjacency to urban settings, formerly agricultural or pre-industrialized sites and the potential for associated identity as well as the potential of forming a network of 10-15km apart.

Tracking the network created in the Ruhr Region through the Revierparks, many characteristics have governed the distribution, the uses, the functions and facilities incorporated in those parks. Such aspects formed a strategic outline to enable the parks to function in the expected roles for development within the areas. The network of Revierparks in the Ruhr took on a linear shape as shown in Fig 11.

The form of the development of the Revierparks into a network resulted through and within the growth directions in the region, synergized with the allocated natural green areas surrounded by the urban settings as a first step. The Revierparks are designated existing green areas within the city that encompass the different above sketched characteristics with potential pre agricultural or pre industrial areas. The concept of this network adopted a spatial approach that balances the relationship between ecological protection and social and economic development and created a sound relationship for a better quality of life on the part of park users. Such an approach enhances and contributes to developing sustainable ecological corridors within the urban tissue [46].

Studying the network of the Revierparks brings benefit to the Amman setting on two fronts. The first is a result of it revealing the ways Amman's green areas could build on and increase the number of their users as well as the frequency of users' visits. Therewith, Amman's green areas could develop an effective layout of green with more initiatives interacting with the surrounding communities. The other beneficial result is in interpreting the Revierparks' approach to situate and validate needed anchor points within the setting of Amman. Those sites

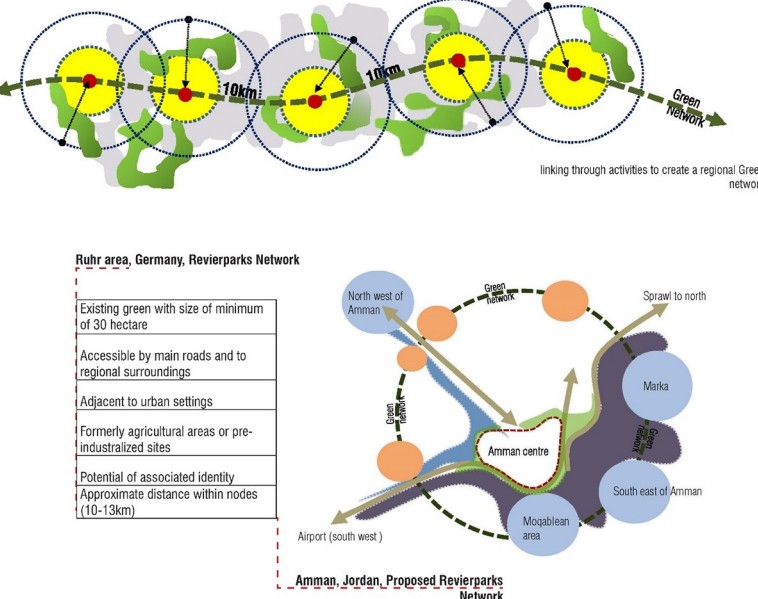

**Fig 11. Revierpark network and possible Amman network.** Source: Authors 2018

designated for such a concept, are selected according to the needs in the different urban areas in Amman. Their role is defined to interrelate with the neighbouring areas through creating different social and economic structures, feeding the area with the services needed but also integrated to have the allocated park within their tissue. The relationship between the parks should take the shape of moving from one center or node to another. These nodes around and within the green would form this network of anchors and clusters expressing life in between and connecting to the other sites through livable corridors of other land uses. This is expected to enhance a better flow of resources within the parts of the city of Amman and therewith, a better quality of life in general. Fig 12 analyses the different layers in adopting this conception for the Amman setting. Many sites have matched the needed size for the conception. Furthermore, the potential of the sites lies in having them accessible to the surrounding local community but also as spaces, that can be developed as entities in their own. In this sense, the idea is to activate the green areas with established centres around them, which are connected to the larger areas through not only green and ecological, but also development corridors that balance the need for environmental protection with the social and economic development. This approach of identifying the developed centres and nodes is needed in Amman to promote the importance of the nodes. If such centres were not well endorsed, the green areas would face danger in their existence in the face of investment construction and further residential sprawl. Therefore, such established nodes and centres around the green areas is thought to be well connected to the other centres by differentiated infrastructure, whether economic, ecological or even transportation networks.

These criteria would consequently increase the centrality of the spaces and the planned open green spaces. Therewith, they would enable orientation of the city of Amman through multiple systems such as transportation and distribution of infrastructural layouts.

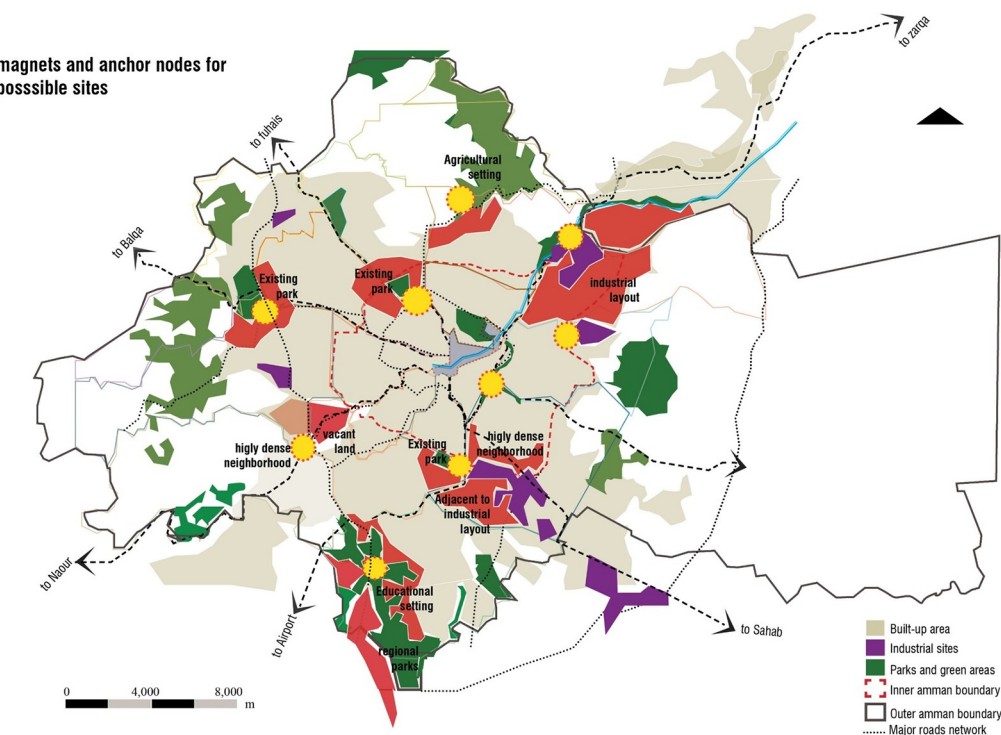

**Fig 12. Green areas in Amman surrounded by urban settings.** Source: Authors 2018

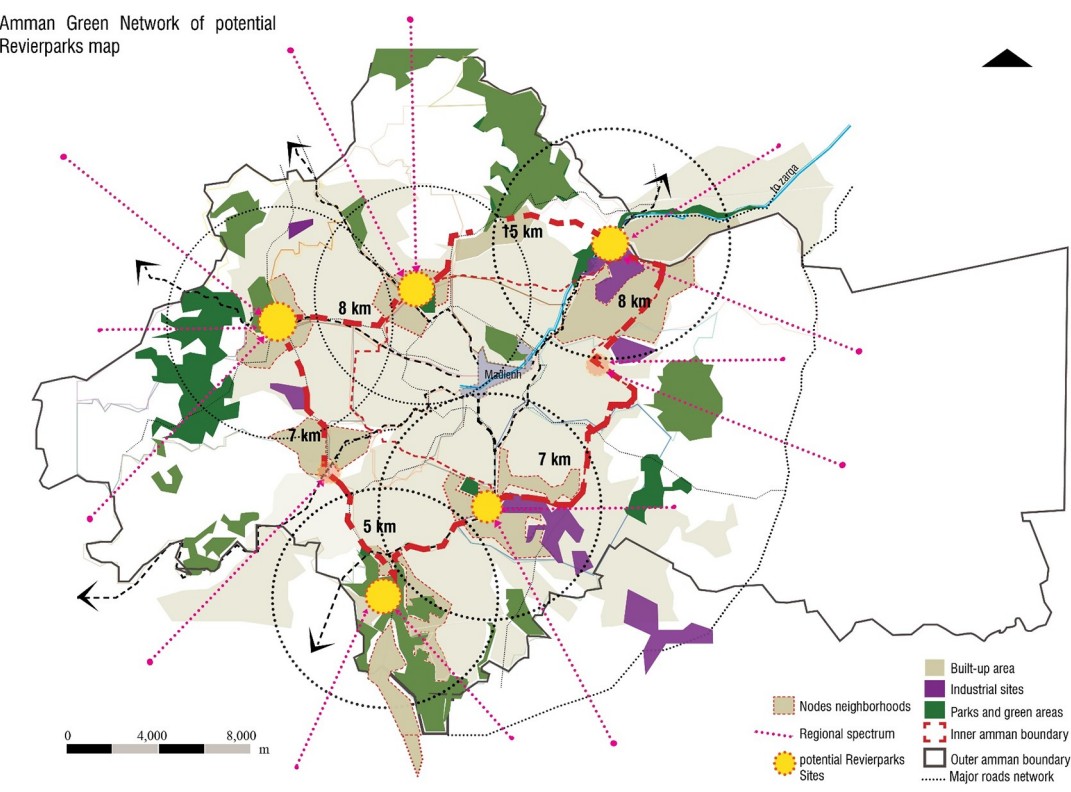

**Fig 13. Resulted network of open green spaces in Amman.** Source: Authors 2018

In this sense, it is also important to identify the planned nodes as sites surrounded by neighbourhoods and urban communities. This would in turn ensure the provision of differentiated activities. It would allow for the envisioning of a potential and a wide framework of associated activities and functions that could cater for multiple social groups visiting the parks rather than having some be excluded due to parks only offering minimal uses related to children's playgrounds and spaces for picnics.

Adapting a Revierpark model for Amman enhances social inclusion on one level and preserves green infrastructure within a green network around the regional layout of Amman on the other. Such nodes of green network play a central role in enabling economic development for the surrounding urban settings.

Further, development of the leftover spaces and settlements in Amman offers a revival opportunity through an organized development. The identity of such spaces and settlements and their prevailing assets are anchor points in reviving an authentic related structure of businesses where functions in these open green spaces are considered. Fig 13 results in a group of consequential nodes that are associated with green, regional layout and which are surrounded by urban settings. These nodes, however, would be optimised to significant locations that are correlated to call attention to the particular characteristics of that area, which would enable them to have their own identity.

Another result of this article relates to the associated unique functions to do with assets found in one particular location rather than the others. As sketched in the discussion above, these assets prioritise functions and programmes only found in these locations, which brings communities and visitors for differentiated reasons to use the area and therewith the open space. Therefore, the node wins an edge on the other ones and brings in activities with other flavours in addition to the common functions of all open green spaces.

## Conclusion

The paper concludes the need for developing Amman through the backbone of the city embodied through the public and green spaces in Amman. The transferability of the integrated concept of the parks through the analysis is seen within the context of Amman.

The conception of the Revierparks sheds light on integrated potential of dealing with the parks. An innovative perception of the parks is tailored through situating existing parks with specific characteristics as hubs for social inclusion, business innovation and centres for the communities. This article results in identifying the needed approach in considering green areas in Amman and integrating them into an interdisciplinary layout that protects the environmental aspects important for the wellbeing of the local community and contributes to a long-term vision for a sustainable development of the city of Amman.

On a strategic level, Amman needs an innovative approach in perceiving development, where a hidden yet related network acts towards social inclusion, better economic development in the declined areas in Amman and a sustainable approach towards preserving green. It is resulted into identifying nodes that correlate between the different users, and act as a catalyst for development in lower income settings as well as other settings through a functional system to each node.

The paper concludes certain recommendations promoting the discussed concept towards multidimensional development. The basic requirement is the provision, protection and management of land by the Amman region. These may be public areas or can be acquired through public private partnerships. It is recommended to have a co-ordinated approach at the regional level that focuses on the social, economic, ecological and regional impacts and highlights the benefits for the residents and actors in the Amman region.

On another level, further study should focus on the drivers energizing and attracting investment in the different selected areas in Amman. Through accompanied levels of services in the identified hubs, synergy can be created to enable holistic development within the network.

## Acknowledgments

Our gratitude goes to German Academic Exchange Service DAAD for funding the Intercultural Dialogue Cooperation Project, Reclaiming Public Space within Metropolitan Regions and facilitating the research among authors. The project was organised by the TU Dortmund, Department of Urban Design and Land Use Planning and the German Jordanian University, School of Architecture and Built Environment.

## Author Contributions

**Conceptualization:** Maram Tawil, Yasemin Utku, Kawthar Alrayyan, Christa Reicher.

**Data curation:** Maram Tawil.

**Formal analysis:** Maram Tawil, Kawthar Alrayyan.

**Investigation:** Maram Tawil, Yasemin Utku, Kawthar Alrayyan.

**Methodology:** Maram Tawil, Yasemin Utku, Kawthar Alrayyan.

**Project administration:** Maram Tawil, Christa Reicher.

**Resources:** Maram Tawil, Yasemin Utku, Kawthar Alrayyan.

**Supervision:** Maram Tawil, Christa Reicher.

**Validation:** Maram Tawil, Christa Reicher.

**Visualization:** Kawthar Alrayyan.

**Writing – original draft:** Maram Tawil, Yasemin Utku, Kawthar Alrayyan.

**Writing – review & editing:** Maram Tawil, Christa Reicher.

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
