## [Decision Letter · Decision Letter 0]

9 Sep 2019

PONE-D-19-20275

Revierparks as an integrated green network in Germany: An option for Amman?

PLOS ONE

Dear Dr. Maram Tawil,

Thank you for submitting your manuscript to PLOS ONE. After careful consideration, we feel that it has merit but does not fully meet PLOS ONE’s publication criteria as it currently stands. Therefore, we invite you to submit a revised version of the manuscript that addresses the points raised during the review process. 

Please note that Reviewer 1 and Reviewer 3 recommends major revisions while Reviewer 2 identifies only minor revisions.We have looked over the comments from three reviewers and find that you should be able to readily accommodate these revisions.

We would appreciate receiving your revised manuscript by 18 October 2019. To enhance the reproducibility of your results, we recommend that if applicable you deposit your laboratory protocols in protocols.io, where a protocol can be assigned its own identifier (DOI) such that it can be cited independently in the future. For instructions see: http://journals.plos.org/plosone/s/submission-guidelines#loc-laboratory-protocols

We look forward to receiving your revised manuscript.

Kind regards,

Eda Ustaoglu, PhD

Academic Editor

PLOS ONE

Journal Requirements:

1. Please ensure that you include a title page within your main document. You should list all authors and all affiliations as per our author instructions and clearly indicate the corresponding author.

Reviewers' comments:

Reviewer's Responses to Questions

**Comments to the Author**

1. Is the manuscript technically sound, and do the data support the conclusions?

Reviewer #1: Partly

Reviewer #2: Yes

Reviewer #3: Partly

2. Has the statistical analysis been performed appropriately and rigorously? 

Reviewer #1: I Don't Know

Reviewer #2: N/A

Reviewer #3: No

3. Have the authors made all data underlying the findings in their manuscript fully available?

Reviewer #1: No

Reviewer #2: Yes

Reviewer #3: No

4. Is the manuscript presented in an intelligible fashion and written in standard English?

Reviewer #1: No

Reviewer #2: Yes

Reviewer #3: Yes

5. Review Comments to the Author

Reviewer #1: Overall, the paper presents a rather interesting topic which can contribute to the relevant literature field. Nevertheless, there are several issues which cannot remain unaddressed if the paper is to be accepted for publication. To that end, I provide the authors with comments which can help them revise their work substantially before submitting it again.

1. The paper must be edited by someone (or copy-editing service) fluent in English as throughout the paper there are major errors in syntax, vocabulary, some typos, grammar, punctuation use, etc. At the same time, the authors must ensure they use the terminology correctly.

2. Regarding the abstract, it does not meet the criteria of an abstract as for the most part it merely reiterates some paragraphs from the Introduction section. Hence, the authors should rewrite their abstract and ensure that their new abstract in one paragraph summarizes the main aspects of the paper encompassing the overall aim of the study and the problem it examines, the key findings that emerged from the analysis as well as a short summary of conclusions based on the findings.

3. In terms of the Introduction section, this should be extensively revised. Although the authors in the Introduction refer to the rationale and context of the study, this is done rather superficially and in hasty manner failing to inform the reader as to what he or she is about to read. Furthermore, the authors make very important statements which are not supported with citations (such as lines 21-22, 27-29, 30-33, 42-43, 49-53). In the revised form of the Introduction, the authors should ensure that they include citations whenever they make an important statement or present information. Regarding the content of the Introduction, the authors should include the relevant information which will inform the reader about the current situation in Amman as well as why the study is important, the problem it tries to solve, and how the study can contribute to the relevant literature strand. Also, in the Introduction (and preferably in the last paragraph of the Introduction) the authors must state explicitly the aim and objectives of the study.

4. The format of citations is in many cases wrong and especially the citations where there are more than two authors (such as the citations in lines 124, 136, 161-162, 167-168, 171-173, 175, 191, 175, 191, 201-202, 205, 227, 281, 382, 394 and son on). I recommend the authors to modify these citations and make sure that all of them follow the appropriate format.

5. The authors should avoid using capital letters in the beginning of words other than names (for example, “Their main” in line 92).

6. Citations should be added in the end of the sentences in lines 76-79, 90-93, 94-96, 96-99, 113-115, 131-135, 136-138, 138-140, 140-144, 144-147, 178-179, 180-181, 181-183, 183-184 since these convey important information.

7. In line 106, after ‘James Mensch’ the year that this work was published must be added in brackets.

8. The main verb in the sentence in lines 109-110 is missing.

9. The sentence “Those activities offered in the field of health and social cohesion” (lines 158-159) cannot stand as it is and should either be integrated in the previous sentence or be written again with the inclusion of some information.

10. The abbreviation and the three punctuation marks (“etc…”) must be deleted in lines 182-183.

11. It is also observed that there are entire paragraphs which do not include citations (such as the paragraph in lines 239-252, 285-313, 317-330, 335-370, 405-412, 413-419, 420-427, 428-431 and so on). Please note that it is not acceptable to provide extensive information without adding references to the sources from which the information was obtained. Citations must be thus added in these paragraphs and anywhere else that information is given.

12. In the section of the Methodology, the authors should clearly describe the sampling method they followed to extract the sample and refer to the size of the sample as well as the month(s) and the year that the survey was performed.

13. The sections of the paper are not well-organized and create confusion. In revising their paper, the authors should ensure that their paper consists of concrete sections which are clearly defined and separated. Moreover, each section should have a title which corresponds to the content of the section. Currently, the theoretical background and the section titled “Amman as a research setting: Methods and data collection” include other sections which create confusion (such as the section ‘Transferability and need oriented analysis’) and the reader does not know whether these sections belong to the main sections or consist separate sections.

14. The authors should add a separate Discussion section in which they will discuss, interpret, compare and contextualize their findings and analysis. Even though they do this to some extent, this should be part of a wider and well-organized Discussion section.

15. Lines 465-479 are the results of the survey and as such they should be presented in a separate section titled “Results”. Thus, the authors should make a separate Results section in which they will thoroughly present their findings by mentioning percentages and other details.

16. The Conclusions section needs a major revision since currently it merely repeats parts from the previous sections without reaching any conclusions. As a result, the Conclusions section is inadequate and looks more like a general and brief discussion. To write their Conclusions effectively, the authors should build on their study findings and analysis to draw meaningful conclusions. Moreover, in the same section they should make recommendations for future research based on their own findings and conclusions.

Reviewer #2: This article uses Armman's revierparks as an example to explore how to build a strategic approach for an open spaces network that encourages dynamic lifestyles.

The article is rich in information and sufficient in discuss, but the content needs to be further focused on specific topics.

My main recommendations of furthur focusing include the following four aspects.

(1) The concept of revierparks is presented by whom and how todefine.

(2) The composition of the armman park system and the role of revierparks.

(3) The universality of armman open space problem, that is, whether revierpark has universal significance

(4) The foothold of this article should not stay in armman itself, but should be extended to the enlightenment and implication

other suggestions are listed

Line34: suggest shape changed to concept

Line92: ""their"" should be capitalized, and there are many similar errors.

Line203: are should be

Line211:wenping L. is a wrong format

Line333: These four kinds of benefits are not the research findings of this article. It is recommended that this section be simplied to one quoted sentence.

Line467: The characteristics of age are obvious young, why

Reviewer #3: Technically the paper needs to be modified by the researcher. Claims regional integration of green spaces is not based on road networks but ecological corridors.

Other comments attached

6. PLOS authors have the option to publish the peer review history of their article (what does this mean?). If published, this will include your full peer review and any attached files.

Reviewer #1: No

Reviewer #2: Yes: Prof. Chen Y.

Reviewer #3: No

---

## [Author Response · Author response to Decision Letter 0]

15 Oct 2019

I have incorporated the reviewers' comments to the best I could. Thank you for the valuable review comments.

The responses to reviewers are in the attachment with the other uploaded files.

Many thanks,

Maram

---

## [Decision Letter · Decision Letter 1]

25 Oct 2019

PONE-D-19-20275R1

Revierparks as an integrated green network in Germany: An option for Amman?

PLOS ONE

Dear Maram Tawil,

Thank you for submitting your manuscript to PLOS ONE. After careful consideration, we feel that it has merit but does not fully meet PLOS ONE’s publication criteria as it currently stands. Therefore, we invite you to submit a revised version of the manuscript that addresses the points raised during the review process.

The reviewers noted that there are still issues with the structure of the paper, discussion and conclusion sections, and citations which require minor changes. The paper should be corrected in terms of grammatical issues and the use of the language.

We would appreciate receiving your revised manuscript by 30 November 2019. To enhance the reproducibility of your results, we recommend that if applicable you deposit your laboratory protocols in protocols.io, where a protocol can be assigned its own identifier (DOI) such that it can be cited independently in the future. For instructions see: http://journals.plos.org/plosone/s/submission-guidelines#loc-laboratory-protocols

We look forward to receiving your revised manuscript.

Kind regards,

Eda Ustaoglu, PhD

Academic Editor

PLOS ONE

Reviewers' comments:

Reviewer's Responses to Questions

**Comments to the Author**

1. If the authors have adequately addressed your comments raised in a previous round of review and you feel that this manuscript is now acceptable for publication, you may indicate that here to bypass the “Comments to the Author” section, enter your conflict of interest statement in the “Confidential to Editor” section, and submit your "Accept" recommendation.

Reviewer #1: (No Response)

Reviewer #2: All comments have been addressed

2. Is the manuscript technically sound, and do the data support the conclusions?

Reviewer #1: Partly

Reviewer #2: Yes

3. Has the statistical analysis been performed appropriately and rigorously? 

Reviewer #1: Yes

Reviewer #2: Yes

4. Have the authors made all data underlying the findings in their manuscript fully available?

Reviewer #1: Yes

Reviewer #2: Yes

5. Is the manuscript presented in an intelligible fashion and written in standard English?

Reviewer #1: No

Reviewer #2: Yes

6. Review Comments to the Author

Reviewer #1: Admittedly the author has to some extent improved the manuscript, however there are still issues to be addressed.

1. Most importantly, the paper needs to be organized in well-defined sections.

2. The Discussion section is still missing and the author should write a Discussion section in which the findings will be interpreted, contextualized and compared to the findings of relevant studies.

3. In addition, citations to support the information given in the Introduction must be added.

4. The Conclusions must be revised extensively since they are somewhat general now. To revise the Conclusions, the authors should draw conclusions based on their research and the points that the research has brought to surface. In this section, they could also refer to the areas a future study should focus on.

5. Finally, even though the English language used in the paper has been improved, there are still errors and minor editing is required. Moreover, there is a moderate tendency to use non-scientific language.

Reviewer #2: The author have made a lot of revesions to the manvscript. However, the auther should repsponse to the reviewer's suggestions one by one.

7. PLOS authors have the option to publish the peer review history of their article (what does this mean?). If published, this will include your full peer review and any attached files.

Reviewer #1: No

Reviewer #2: Yes: Yiyong CHEN

---

## [Author Response · Author response to Decision Letter 1]

9 Nov 2019

1 The paper needs to be organized in well-defined sections 

We went through the structure of the paper and modified in the headings and sections to have it more structured upon the comment of the reviewer

2 The Discussion section is still missing and the author should write a Discussion section in which the findings will be interpreted, contextualized and compared to the findings of relevant studies. 

The section is added and interpreted as required to have the findings contextualized as far as possible

Please see pages 24-30

3 Citations to support the information given in the Introduction must be added. 

Citations were added to support the introduction. Please see pages 2-3

4 The Conclusions must be revised extensively since they are somewhat general now. To revise the Conclusions, the authors should draw conclusions based on their research and the points that the research has brought to surface. In this section, they could also refer to the areas a future study should focus on 

The conclusion was revised and we tried to build on what the research has revealed and resulted in. Future areas of study were also highlighted.

Please see pages 33-35

5 Finally, even though the English language used in the paper has been improved, there are still errors and minor editing is required. Moreover, there is a moderate tendency to use non-scientific language 

The paper was thoroughly proof read and edited. Editing can be found in all pages of the paper

---

## [Decision Letter · Decision Letter 2]

25 Nov 2019

PONE-D-19-20275R2

Revierparks as an integrated green network in Germany: An option for Amman?

PLOS ONE

Dear Dr Maram Tawil,

Thank you for submitting your manuscript to PLOS ONE. After careful consideration, we feel that it has merit but does not fully meet PLOS ONE’s publication criteria as it currently stands. Therefore, we invite you to submit a revised version of the manuscript that addresses the points raised by Reviewer 1 during the review process.

We would appreciate receiving your revised manuscript by 23 December 2019. To enhance the reproducibility of your results, we recommend that if applicable you deposit your laboratory protocols in protocols.io, where a protocol can be assigned its own identifier (DOI) such that it can be cited independently in the future. For instructions see: http://journals.plos.org/plosone/s/submission-guidelines#loc-laboratory-protocols

We look forward to receiving your revised manuscript.

Kind regards,

Eda Ustaoglu, PhD

Academic Editor

PLOS ONE

Reviewers' comments:

Reviewer's Responses to Questions

**Comments to the Author**

1. If the authors have adequately addressed your comments raised in a previous round of review and you feel that this manuscript is now acceptable for publication, you may indicate that here to bypass the “Comments to the Author” section, enter your conflict of interest statement in the “Confidential to Editor” section, and submit your "Accept" recommendation.

Reviewer #1: (No Response)

Reviewer #2: All comments have been addressed

2. Is the manuscript technically sound, and do the data support the conclusions?

Reviewer #1: Partly

Reviewer #2: Yes

3. Has the statistical analysis been performed appropriately and rigorously? 

Reviewer #1: Yes

Reviewer #2: Yes

4. Have the authors made all data underlying the findings in their manuscript fully available?

Reviewer #1: Yes

Reviewer #2: Yes

5. Is the manuscript presented in an intelligible fashion and written in standard English?

Reviewer #1: Yes

Reviewer #2: Yes

6. Review Comments to the Author

Reviewer #1: The authors have indeed improved many aspects of the paper, especially, in terms of the English language. The overall organization of the paper has also been improved. Hence, I encourage the authors to perform only the following changes:

1. Pay attention to the titles of the sections. For example, "Introduction and focus of the investigation" as well as "Discussion and analysis" are not appropriate and should be replaced with "Introduction" and "Discussion", respectively.

2. In the last paragraph of the Introduction, the authors should clearly state the objectives of the papers (as they have done in lines 479-481 on page 21).

3. To have a methodologically robust research paper, it is necessary that the authors describe the methodology they followed to carry out the survey. In specific, it is very important to describe the sampling method as well as to explain how the sample size was determined. Moreover, the authors should add some information about the queastionnaire and, in specific, refer to the literature sources they used to design the questionnaire. Moreover, they could also refer to the content and answer scales of the questionnaire.

4. Regarding the Discussion, although the results are discussed, this is largely done without referring to previous relevant research works. To put this differently, the authors should state whether their study results confirm or contradict previous literature works which have been presented in the theoretical parts of their paper.

Reviewer #2: All my main concerns have been addressed in the revised manuscript. I would like to recommend its publication.

7. PLOS authors have the option to publish the peer review history of their article (what does this mean?). If published, this will include your full peer review and any attached files.

Reviewer #1: No

Reviewer #2: No

---

## [Author Response · Author response to Decision Letter 2]

3 Dec 2019

Thank you for continuous support and valid comments to raise the quality of the paper. Below, please find the response to the reviewers’ comments:

1 Pay attention to the titles of the sections. For example, "Introduction and focus of the investigation" as well as "Discussion and analysis" are not appropriate and should be replaced with "Introduction" and "Discussion", respectively. 

The titles were replaced as recommended. Please see pages 2 and 25

2 In the last paragraph of the Introduction, the authors should clearly state the objectives of the papers (as they have done in lines 479-481 on page 21) 

The objectives were stated in the last paragraph of the introduction as required in your comment. Please see page 4

3 To have a methodologically robust research paper, it is necessary that the authors describe the methodology they followed to carry out the survey. In specific, it is very important to describe the sampling method as well as to explain how the sample size was determined. Moreover, the authors should add some information about the questionnaire and, in specific, refer to the literature sources they used to design the questionnaire. Moreover, they could also refer to the content and answer scales of the questionnaire.

 The methodology followed along with a description of the sampling and the literature reference to choosing the method were done according to the comment. Thank you for guiding the paper to be in a better shape. The modifications were done in pages 22 and 23.

4 Regarding the Discussion, although the results are discussed, this is largely done without referring to previous relevant research works. To put this differently, the authors should state whether their study results confirm or contradict previous literature works which have been presented in the theoretical parts of their paper. 

According to the reviewer’s comment, the reference to confirming or contradicting previous literature works has been done to state the results within the context and framework of discussion. Modified parts are found in pages 25-28

---

## [Editor Report · Decision Letter 3]

9 Dec 2019

Revierparks as an integrated green network in Germany: An option for Amman?

PONE-D-19-20275R3

Dear Dr. Tawil,

We are pleased to inform you that your manuscript has been judged scientifically suitable for publication and will be formally accepted for publication once it complies with all outstanding technical requirements.

With kind regards,

Eda Ustaoglu, PhD

Academic Editor

PLOS ONE
---

## [Editor Report · Acceptance letter]

12 Dec 2019

PONE-D-19-20275R3 

Revierparks as an integrated green network in Germany: An option for Amman? 

Dear Dr. Tawil:

I am pleased to inform you that your manuscript has been deemed suitable for publication in PLOS ONE. Congratulations! Your manuscript is now with our production department. 

With kind regards,

on behalf of

Dr. Eda Ustaoglu 

Academic Editor

PLOS ONE